# Nonlinear Dynamic Response of Nanocomposite Microbeams Array for Multiple Mass Sensing

**DOI:** 10.3390/nano13111808

**Published:** 2023-06-05

**Authors:** Giovanni Formica, Walter Lacarbonara, Hiroshi Yabuno

**Affiliations:** 1Department of Architecture, Roma Tre University, 33328 Rome, Italy; 2Department of Structural and Geotechnical Engineering, Sapienza University of Rome, 33328 Rome, Italy; walter.lacarbonara@uniroma1.it; 3Faculty of Engineering, Information and Systems, University of Tsukuba, Tsukuba 300-4352, Japan; yabuno@esys.tsukuba.ac.jp

**Keywords:** nanocomposite, microcantilever, mass sensing, frequency shifts, carbon nanotubes, nonlinear frequency response

## Abstract

A nonlinear MEMS multimass sensor is numerically investigated, designed as a single input-single output (SISO) system consisting of an array of nonlinear microcantilevers clamped to a shuttle mass which, in turn, is constrained by a linear spring and a dashpot. The microcantilevers are made of a nanostructured material, a polymeric hosting matrix reinforced by aligned carbon nanotubes (CNT). The linear as well as the nonlinear detection capabilities of the device are explored by computing the shifts of the frequency response peaks caused by the mass deposition onto one or more microcantilever tips. The frequency response curves of the device are obtained by a pathfollowing algorithm applied to the reduced-order model of the system. The microcantilevers are described by a nonlinear Euler-Bernoulli inextensible beam theory, which is enriched by a meso-scale constitutive law of the nanocomposite. In particular, the microcantilever constitutive law depends on the CNT volume fraction suitably used for each cantilever to tune the frequency bandwidth of the whole device. Through an extensive numerical campaign, the mass sensor sensitivity estimated in the linear and nonlinear dynamic range shows that, for relatively large displacements, the accuracy of the added mass detectability can be improved due to the larger nonlinear frequency shifts at resonance (up to 12%).

## 1. Introduction

In the past decades MEMS have been subject to disruptive developments in several fields of engineering both in research and applications (mainly, signal processing, actuation, and sensing). The lower power consumption and higher sensitivity are distinct advantages with respect to comparable macro devices. A general introduction about MEMS devices can be found in [1] with an overview of the fundamental concepts pertaining to linear and nonlinear dynamics of such systems. In recent years, a great deal of studies have sought to advance our understanding of the nonlinear dynamic behavior of nano/microresonators, which represent the main components of actual NEMS/MEMS devices [2,3,4,5]. Nonlinear dynamic aspects of electrostatically actuated microcantilevers are discussed in [6] while dynamics of the same systems subject to piezoelectric actuation are studied in [7].

Accurate models still need to achieve better agreement between analytical/numerical predictions and experiment results, and explore the use of new materials and technical capabilities. Initial imperfections of the structure, together with electrostatic actuation and nonlinear damping effects, are discussed in [8,9]. Moreover, systems with feedback loop control are investigated to enhance the sensitivity of the microdevices [10,11]. By restricting the attention to micromass sensing, the literature is becoming increasingly richer due to the present interest in gas detection, chemical agent detection, biological molecules and pollutant particles identification. In mass/gas detection, different physical principles can be used to measure the different target substances. In [12,13,14] some electrical resistive detectors are investigated. In these studies the functioning of a semiconductor placed in an integrated circuit aimed to measure target molecules such as SO${}_{2}$ or NO${}_{2}$ in order to perform environmental control or to measure the pollution associated with combustion chamber exhaust gases. Moreover, an example of a MEMS gas sensor using Knudsen force for hydrogen detection can be found in [15,16]. Therefore, MEMS gas sensors using the mechanical features of micro oscillators are of current interest. An overview of the working principles of these devices can be found in [17]. A micro mass sensor based on a microcantilever is discussed in [18]. This micromechanical resonator is used as gas sensor exploiting a working principle based on the detection of the beam natural frequency shift due to mass deposition onto the tip of the microbeam. In addition, similar works addressing microcantilevers used as sensor resonators to detect the chemical concentration of matter are found in [19,20]. On the other hand, a detection procedure is discussed in [21,22] where monitoring of the amplitudes of the localized vibration modes that vary when the system mass is perturbed was performed. A device made of two microcantilevers weakly coupled through an overhang working with localized modes is discussed in [23] while the use of these tools in highly viscous fluids is explored in [24,25]. Different kinds of electrostatic actuation are addressed in [26,27] to achieve large motions as well as higher sensitivity. In the field of NEMS, the performance of standalone carbon nanotube oscillators for mass detection is investigated in [28,29].

Numerous studies have been carried out to explore new mass detection strategies exploiting nonlinear effects. A parametrically excited clamped-clamped beam with cubic stiffness nonlinearity is explored as micromass sensor in [30]. The principle is based on the passage from the stable region to the unstable region across the transition curves of the Strutt diagrams. Also the jumps between the nonresonant and resonant branches occurring at the fold bifurcations along the frequency response curves are used as detection principle in [31,32]. Moreover, an electrostatically actuated microcantilever dynamic mass sensor was analytically investigated and experimentally tested in [33] to indirectly measure the added mass and its position along the span exploiting the frequency shift. The nonlinear dynamic behavior of an electrostatically actuated microbeam was studied in [34] to compare the different kinds of micromass detections according to the jumps at the bifurcation points, the frequency sweeps, and the frequency shifts of the resonant peaks. A specific study on the static and dynamic effects of the pull-in phenomena for a MEMS gas sensor can be found in [35]. For an extensive and up-to-date review, the reader can refer to [36], where a comprehensive categorization of micro/nanomechanical resonators and their engineering applications is provided, including a classification of NEMS/MEMS devices based on linear/nonlinear, single/array, symmetric/asymmetric, together with frequency shift-based and amplitude shift-based resonators.

In the present work, a single-input single-output (SISO) system is investigated making use of a path following algorithm optimized to tackle several degrees of freedom. The baseline idea of the SISO system can be found in [37] where a dynamic lumped system is fabricated to work in the linear range, and then successfully experimented for detection purposes, with experimental results in good agreement with a previous theoretical contribution of the same author [38]. The working principle of the device consists in the use of the shuttle mass motion both to excite the sensors and to read off the overall dynamic response of the system which includes the vibrations of all microcantilevers.

One of the key aspects of the proposed design is the use of CNT nanocomposite material (i.e., polymeric matrix hosting aligned CNTs forests) for the microcantilever detectors. Due to their excellent mechanical, thermal, electrical, and tribological properties, compared with their lightweight nature, CNT-based nanocomposites reached in the last decades a widespread use in various engineering applications (see, e.g., [39,40,41,42,43] specifically for reinforced polymers). For sensor applications, their ability of specific and stable detections of mechanical and electrochemical properties was extensively explored in the literature: for an up-to-date survey on this, see for instance [44,45]. An experimental and numerical study about the response of such nanostructured materials is given in [46,47,48], where the accuracy of the constitutive modeling here employed was also successfully validated.

On the other hand, the analytical investigation via a perturbation technique of the design guidelines for SISO systems can be found in [49]. In the present work a different approach based on the eigenvalue analysis for a five-dof system is presented. Moreover, contrary to the conventional way of tuning the detector frequencies by changing their length, the proposed design consists of an array of equal-length beams having different CNT volume fractions which leads to easily tunable frequencies. The shuttle mass is a mass-spring-damper system while the microcantilevers clamped to the shuttle mass are modeled as nonlinear Euler-Bernoulli beams, enhanced by the constitutive relationship derived at meso-scale level for CNT-nanocomposite beams as proposed in [50]. The equations of motion are discretized via the Galerkin method and the system sensitivity is investigated for different cases such as the case of the system driven by a harmonic excitation or by an electrostatic actuation as well as the case of the system with one or more working sensor beams. Finally, the sensitivity of the device is computed in the linear and nonlinear dynamic ranges proving that the nonlinearities can enhance the detection capabilities.

## 2. Equations of Motion for the Nanocomposite Microcantilevers

The SISO system investigated in the present study is depicted in  Figure 1.

The flexural motion of the *j*th microcantilever is described by the nonlinear Euler-Bernoulli beam theory: for full details, we refer the interested reader to [51]. Let *s* and *t* be the arclength along the beam baseline and time, respectively, and let $\partial_{s}$ and $\partial_{t}$ denote differentiation with respect to *s* and *t*, respectively. The nonlinear equation of motion for the *j*th undamped microcantilever is given by the following integral-partial differential equation:(1)$$\begin{array}{cc} \; & \rho A\left(\partial_{tt}v_{j}\cos\theta_{j}-\partial_{tt}u_{j}\sin\theta_{j}\right)-\kappa_{j}\;N_{j}^{*}+\partial_{ss}M_{j}\\ \; & +\kappa_{j}\int_{s}^{l}\left[\rho A\left(\partial_{tt}u_{j}\cos\theta_{j}+\partial_{tt}v_{j}\sin\theta_{j}\right)-\kappa_{j}\partial_{s}M_{j}\right]d\xi =0\;, \end{array}$$
where $N_{j}^{*}$ is the normal force at the tip of the cantilever, which is due to the presence of the added mass at the tip denoted by $m_{j}$; ρA is the beam mass per unit length, with *l* being its span, while $u_{j}$ and $v_{j}$ are the axial and transverse (i.e., with respect to the fixed directions $e_{3}$ and $e_{1}$) displacements of the cantilevers, respectively. Furthermore, $\theta_{j}$ denotes the counterclockwise rotation of the beam cross section, $\kappa_{j}$ is the flexural curvature, and $M_{j}=EJ_{j}^{\mathrm{eq}}\kappa_{j}$ is the elastic bending moment, proportional to the equivalent flexural stiffness $EJ_{j}^{\mathrm{eq}}$ of the *j*th nanocomposite cantilever cross section. In particular, according to [47,50,52,53], such equivalent flexural stiffness is derived by a 3D continuum homogeneization model which provides the following expression:(2)$$EJ_{j}^{\mathrm{eq}}={\hat{EJ}}^{\mathrm{eq}}\left(\phi_{C},\;E_{M},\;E_{C},\;\nu_{M},\;\nu_{C}\right)\;,$$
where the equivalent stiffness depends on the volume fraction $\phi_{C}$ of the CNTs embedded in the polymeric hosting matrix, and of the isotropic elastic moduli of the two materials, Young’s and Poisson’s moduli $E_{C}$ and $\nu_{C}$, respectively for the CNTS, Young’s and Poisson’s moduli $E_{M}$ and $\nu_{M}$, respectively for the hosting polymeric matrix.

The boundary conditions for the *j*th microcantiliver considering the presence of the added mass $m_{j}$ at its tip are:(3)$$\begin{array}{cc} \; & v_{j}\left(0,t\right)=x\left(t\right),\;\partial_{s}v_{j}\left(0,t\right)=0,\;\partial_{ss}v_{j}\left(l,t\right)=0,\\ & EJ_{j}^{\mathrm{eq}}\partial_{sss}v_{j}\left(l,t\right)=m_{j}\partial_{tt}v_{j}\left(l,t\right)\;. \end{array}$$

Note that the dynamic boundary condition involving the tip mass $m_{j}$ is obtained via linearization of the fully nonlinear equation of motion for the lumped tip mass. Such equation enforces the balance of linear momentum at the cantilever tip expressed as
(4)$$\left(N_{j}^{*}b_{3}+Q_{j}^{*}b_{1}\right)=-m_{j}\left[\partial_{tt}u_{j}\left(l,\;t\right)e_{3}+\partial_{tt}v_{j}\left(l,\;t\right)e_{1}\right].$$
where $\left(b_{1},b_{3}\right)$ are the beam cross section-fixed unit vectors providing the current cross section orientation and the direction normal to it while $\left(e_{1},e_{3}\right)$ indicate the fixed horizontal and vertical directions, respectively. Therefore, the *j*th tension $N_{j}^{*}$ and shear force $Q_{j}^{*}$ can be obtained as
(5)$$\begin{array}{c} N_{j}^{*}=-m_{j}\left[\cos\theta_{j}\left(l,\;t\right)\;\partial_{tt}u_{j}\left(l,\;t\right)+\sin\theta_{j}\left(l,\;t\right)\;\partial_{tt}v_{j}\left(l,\;t\right)\right], \end{array}$$
(6)$$\begin{array}{c} Q_{j}^{*}=-m_{j}\left[-\sin\theta_{j}\left(l,\;t\right)\;\partial_{tt}u_{j}\left(l,\;t\right)+\cos\theta_{j}\left(l,\;t\right)\;\partial_{tt}v_{j}\left(l,\;t\right)\right]. \end{array}$$

The subsequent linearization of Equation (Equation 6) yields
(7)$$Q_{j}^{*}=-m_{j}\partial_{tt}v_{j}\left(l,t\right).$$

By further expanding Equation (Equation 1) in Taylor series up to the third order adopting the moment-curvature law proposed in [52,53], and further developed in [50], expressing the longitudinal motion $u_{j}$ as a function of the transverse motion $v_{j}$, the Euler-Bernoulli equations of motion governing the transverse dynamics of the *j*th microbeam with the tip mass $m_{j}$ are cast in the form
(8)$$\begin{array}{cc} \; & \rho A\partial_{tt}v_{j}\;\left(1-\frac{1}{2}{\left(\partial_{s}v_{j}\right)}^{2}\right)+\rho A\partial_{s}v_{j}\int_{0}^{s}\left[\partial_{s}v_{j}\partial_{stt}v_{j}+{\left(\partial_{st}v_{j}\right)}^{2}\right]d\xi \\ \; & +\partial_{ss}v_{j}\;\int_{s}^{l}\rho A\;\left[\partial_{s}v_{j}\partial_{tt}v_{j}-\int_{0}^{\xi }\left(\partial_{s}v_{j}\partial_{stt}v_{j}+{\left(\partial_{st}v_{j}\right)}^{2}\right)dz\right]d\xi \\ \; & +\partial_{ss}M_{j}-\partial_{ss}v_{j}\int_{s}^{l}\partial_{ss}v_{j}\partial_{s}M_{j}\;d\xi \\ \; & +m_{j}\partial_{ss}v_{j}\left[\left(\partial_{s}v_{j}\right){|}_{l}\left(\partial_{tt}v_{j}\right){|}_{l}+\int_{0}^{l}\left(\partial_{s}v_{j}\partial_{stt}v_{j}+{\left(\partial_{st}v_{j}\right)}^{2}\right)d\xi \right]=0\;, \end{array}$$
where $v_{j}$ is the transverse deflection of the beam, the constitutive law for the bending moment is
$$M_{j}=EJ_{j}^{\mathrm{eq}}\kappa_{j}=EJ_{j}^{\mathrm{eq}}\left[\partial_{ss}v_{j}\left(1+\frac{1}{2}{\left(\partial_{s}v_{j}\right)}^{2}\right)\right]\;\mathrm{with}\;\;\kappa_{j}=\partial_{ss}v_{j}\left(1+\frac{1}{2}{\left(\partial_{s}v_{j}\right)}^{2}\right)\;.$$

Finally, the equation of motion for the shuttle mass is given by
(9)$$M\partial_{tt}x+c\partial_{t}x+Kx+\sum_{j=1}^{N_{b}}\partial_{s}M_{j}{|}_{s=0}=f\cos\Omega t\;,$$
where *x* is the displacement of the shuttle mass *M*, *c* is the damping coefficient of the dashpot, *k* the spring constant, f(t)=FcosΩt with *F* and Ω being the excitation amplitude and frequency, respectively. The last term is the sum the shear forces of the $N_{b}$ beams at the roots attached to the shuttle mass, on accounting for the relationship between shear force and moment gradient $Q=-\partial_{s}M$. In fact, the associated boundary conditions for the beams are the kinematic relations for the clamp and the mechanical boundary conditions at the tip
$$v_{j}\left(0,t\right)=\partial_{s}v_{j}\left(0,t\right)=x\left(t\right)=0$$
and
$$M_{j}\left(l,t\right)=0,\;Q_{j}\left(l,t\right)=-m_{j}\left(\partial_{tt}x+\partial_{tt}v_{j}\right){|}_{s=l}\;.$$

Expressed in terms of displacements, the mechanical the boundary conditions at the tip become
$$EJ_{j}^{\mathrm{eq}}\partial_{ss}v_{j}\left(l,t\right)=0\;,\;EJ_{j}^{\mathrm{eq}}\partial_{sss}v_{j}\left(l,t\right)=m_{j}\left(\partial_{tt}x+\partial_{tt}v_{j}\right){|}_{s=l}\;.$$

We rescale the absolute displacements $v_{j}$ of the *j*th beam and the shuttle mass displacement *x* by the length of the beams *l*, time by $1/\omega_{c}$ with $\omega_{c}^{2}=EJ^{\mathrm{eq}}/\left(\rho Al^{4}\right)$, where ρA is the mass per unit length of the reference beam whose bending stiffness is denoted by $EJ^{\mathrm{eq}}$. By retaining the symbols *s* and *t* as the beam nondimensional arclength and time, respectively, and using the prime for space derivatives and the over dot for time derivatives, the ensuing nondimensional Equation (Equation 8) become:(10)$$\begin{array}{cc} \; & \mu \;\ddot{x}\left(t\right)+\xi \;\dot{x}\left(t\right)+K^{*}\;x\left(t\right)+\sum_{j=1}^{N_{b}}\alpha_{j}v_{j}^{\prime \prime \prime }\left(0,t\right)=f^{*}\;\cos\Omega^{*}t\;,\\ \; & \mu_{j}\;{\ddot{v}}_{j}\left(s,t\right)+m_{j}^{*}\;\delta \left(s-1\right){\ddot{v}}_{j}\left(s,t\right)+\xi_{j}\;{\dot{v}}_{j}\left(s,t\right)+\alpha_{j}v_{j}^{\prime \prime \prime \prime }\left(s,t\right)\\ \; & +\mu_{j}\;f_{j}^{I}\left[v\left(s,t\right),\;\dot{v}\left(s,t\right),\;\ddot{v}\left(s,t\right)\right]+\alpha_{j}\;f_{j}^{R}\left[v\left(s,t\right)\right]\\ \; & +m_{j}^{*}\;f_{j}^{I*}\left[v\left(s,t\right),\;\dot{v}\left(s,t\right),\;\ddot{v}\left(s,t\right)\right]=0\;,\;\mathrm{for}\;j=1,\;\ldots ,\;N_{b}\;, \end{array}$$
where
$$\begin{array}{cc} f_{j}^{I}\left[v,\;\dot{v},\;\ddot{v}\right] & =-\frac{1}{2}{\ddot{v}}_{j}{\left(v_{j}^{\prime }\right)}^{2}+v_{j}^{\prime \prime }\int_{s^{★}}^{1}\left[{\ddot{v}}_{j}\;v_{j}^{\prime }\right]d\xi^{★}+v_{j}^{\prime }\int_{0}^{s^{★}}\left[v_{j}^{\prime }{\ddot{v}}_{j}^{\prime }\right]d\xi^{★}\\ & -v_{j}^{\prime \prime }\int_{s^{★}}^{1}\left[\int_{0}^{s^{★}}\left[v_{j}^{\prime }{\ddot{v}}_{j}^{\prime }\right]d\xi^{★}\right]ds^{★}+v_{j}^{\prime }\int_{0}^{s^{★}}\left[{\left({\dot{v}}_{j}^{\prime }\right)}^{2}\right]d\xi^{★}\\ & -v_{j}^{\prime \prime }\int_{s^{★}}^{1}\left[\int_{0}^{s^{★}}\left[{\left({\dot{v}}_{j}^{\prime }\right)}^{2}\right]d\xi^{★}\right]ds^{★}\;,\\ f_{j}^{R}\left[v\right] & ={\left(v_{j}^{\prime \prime }\right)}^{3}+\frac{1}{2}v_{j}^{\prime \prime \prime \prime }{\left(v_{j}^{\prime }\right)}^{2}+3v_{j}^{\prime \prime \prime }v_{j}^{\prime \prime }v_{j}^{\prime }-v_{j}^{\prime \prime }\int_{s^{★}}^{1}\left(v_{j}^{\prime \prime }v_{j}^{\prime \prime \prime }+\ldots \right)d\xi^{★}\;,\\ f_{j}^{I*}\left[v,\;\dot{v},\;\ddot{v}\right] & =v_{j}^{\prime \prime }\left[\left(v_{j}^{\prime }\right){|}_{1}\left({\ddot{v}}_{j}\right){|}_{1}-\int_{0}^{1}\left(v_{j}^{\prime }{\ddot{v}}_{j}^{\prime }+{\left({\dot{v}}_{j}^{\prime }\right)}^{2}\right)d\xi \right]\;. \end{array}$$

In the governing equations, we introduced the following nondimensional parameters:$$\begin{array}{c} \mu :=M/\left(\rho Al\right)\;,\;\mu_{j}:=\rho A_{j}/\rho A\;,\;m_{j}^{*}:=m_{j}/\left(\rho Al\right)\;,\\ \xi :=c/\left(\rho Al\omega_{c}\right),\;\;\xi_{j}:=c_{j}/\left(\rho A\omega_{c}\right)\;,\\ K^{*}:=K/\left(\rho Al\omega_{c}^{2}\right)\;,\;\alpha_{j}:=EJ_{j}^{\mathrm{eq}}/\left(\rho A\omega_{c}^{2}l^{4}\right)\;,\\ f^{*}:=f/\left(\rho Al^{2}\omega_{c}^{2}\right)\;,\;\Omega^{*}:=\Omega /\omega_{c}\;, \end{array}$$
where $m_{j}$ is the tip mass of the *j*th beam, $\xi_{j}$ its damping coefficient, (μ, $K^{*}$, ξ) are the nondimensional mass, spring constant and damping coefficient of the shuttle mass, $f^{*}$ and $\Omega^{*}$ are the nondimensional excitation amplitude and frequency, respectively.

We further introduce the beam deflections $w_{j}$ relative to the shuttle mass according to $v_{j}\left(s,t\right)=w_{j}\left(s,t\right)+x\left(t\right)$. The equations of motion in terms of the relative beam deflections $w_{j}$ take the following form:(11)$$\begin{array}{cc} \; & \mu \;\ddot{x}\left(t\right)+\xi \;\dot{x}\left(t\right)+K^{*}\;x\left(t\right)+\sum_{j=1}^{N_{b}}\alpha_{j}w_{j}^{\prime \prime \prime }\left(0,t\right)=f^{*}\;\cos\Omega^{*}t\;,\\ \; & \mu_{j}\;\left({\ddot{w}}_{j}\left(s,t\right)+\ddot{x}\left(t\right)\right)+m_{j}^{*}\;\delta \left(s-1\right)\;\left({\ddot{w}}_{j}\left(s,t\right)+\ddot{x}\left(t\right)\right)+\xi_{j}\;{\dot{w}}_{j}\left(s,t\right)+\alpha_{j}w_{j}^{\prime \prime \prime \prime }\left(s,t\right)\\ & +\alpha_{j}\;f_{j}^{R}\left[w\left(s,t\right)\right]+\mu_{j}\;f_{j}^{I}\left[w\left(s,t\right),\;\dot{w}\left(s,t\right),\;\ddot{w}\left(s,t\right)\right]+m_{j}^{*}\;f_{j}^{I*}\left[w\left(s,t\right),\;\dot{w}\left(s,t\right),\;\ddot{w}\left(s,t\right)\right]\\ & +\mu_{j}\;g_{j}^{I}\left[w\left(s,t\right)\right]\;\ddot{x}\left(t\right)+m_{j}^{*}\;g_{j}^{I*}\left[w\left(s,t\right)\right]\;\ddot{x}\left(t\right)=0\;,\;\mathrm{for}\;j=1,\;\ldots ,\;N_{b}\;, \end{array}$$
where
$$\begin{array}{c} g_{j}^{I}\left[w\right]=-\frac{1}{2}{\left(w_{j}^{\prime }\right)}^{2}+w_{j}^{\prime \prime }\int_{s^{★}}^{1}\left[w_{j}^{\prime }\right]d\xi^{★}\;,\;g_{j}^{I*}\left[w\right]=w_{j}^{\prime \prime }\left(w_{j}^{\prime }\right){|}_{1}\;. \end{array}$$

## 3. Modal Analysis

The modal analysis of the SISO system here discussed is the first step towards the nonlinear analyses carried out on the Galerkin obtained reduced order models. Moreover, the variability of the eigenfrequencies of the SISO system reveals interesting preliminary results, thanks to the nanostructured nature of the materials constituting the microbeams.

Let the *j*th overall mode shape of the system be cast in the vector-valued form:(12)$$u_{j}^{⊤}=\left[X_{j},\;\phi_{1,j}\left(s\right),\;\phi_{2,j}\left(s\right),\;\ldots ,\;\phi_{N_{b},j}\left(s\right)\right]=\left[X_{j},\;\phi_{k,j}\left(s\right)\right]\;,\;k=1,\ldots N_{b}\;.$$

For further details see Appendix A. The substitution of the *j*th mode in the nondimensional *k*th beam equation and shuttle mass equation yields
(13)$$\begin{array}{c} -\mu_{k}\omega_{j}^{2}\phi_{k,j}+\alpha_{k}\phi_{k,j}^{\prime \prime \prime \prime }=0\;, \end{array}$$
(14)$$\begin{array}{c} -\mu \omega_{j}^{2}X_{j}+K^{*}X_{j}+\sum_{k}\alpha_{k}\phi_{k,j}^{\prime \prime \prime }\left(0\right)=0\;; \end{array}$$
the associated boundary conditions for the beams are
(15)$$\begin{array}{c} \phi_{k,j}\left(0\right)-X_{j}=0\;,\;\phi_{k,j}^{\prime }\left(0\right)=0\;, \end{array}$$
(16)$$\begin{array}{c} \alpha_{k}\phi_{k,j}^{\prime \prime }\left(1\right)=0\;,\;\alpha_{k}\phi_{k,j}^{\prime \prime \prime }\left(1\right)+m_{k}^{*}\omega_{j}^{2}\phi_{k,j}\left(1\right)=0\;. \end{array}$$

We multiply Equation (Equation 13) by $\phi_{k,i}$, integrate over [0,1] and sum the resulting projected equations for all beams; we multiply Equation (Equation 14) by $X_{i}$ and sum the previously projected beam equations and the shuttle mass equation to obtain:(17)$$\begin{array}{cc} -\omega_{j}^{2}[\sum_{k=1}^{N_{b}} & \mu_{k}\int_{0}^{1}\phi_{k,i}\phi_{k,j}ds+\mu X_{i}X_{j}]+\sum_{k=1}^{N_{b}}\alpha_{k}\int_{0}^{1}\phi_{k,i}\phi_{k,j}^{\prime \prime \prime \prime }ds\\ \; & +K^{*}X_{i}X_{j}+X_{i}\sum_{k=1}^{N_{b}}\alpha_{k}\phi_{k,j}^{\prime \prime \prime }\left(0\right)\;.=0 \end{array}$$

We further integrate by parts the bending terms in the beam equations to obtain
(18)$$\int_{0}^{1}\phi_{k,i}\phi_{k,j}^{\prime \prime \prime \prime }ds=\phi_{k,i}\phi_{k,j}^{\prime \prime \prime }{{|}_{s=0}^{1}-\phi_{k,i}^{\prime }\phi_{k,j}^{\prime \prime }|}_{s=0}^{1}+\int_{0}^{1}\phi_{k,i}^{\prime \prime }\phi_{k,j}^{\prime \prime }ds\;.$$

Employing the boundary conditions (Equation 15) yields
(19)$$\int_{0}^{1}\phi_{k,i}\phi_{k,j}^{\prime \prime \prime \prime }ds=-\phi_{k,i}\left(1\right)\omega_{j}^{2}\frac{m_{k}^{*}}{\alpha_{k}}\phi_{k,j}\left(1\right)-X_{i}\phi_{k,j}^{\prime \prime \prime }\left(0\right)+\int_{0}^{1}\phi_{k,i}^{\prime \prime }\phi_{k,j}^{\prime \prime }ds\;.$$

Substituting (Equation 19) into (Equation 20) yields
(20)$$\begin{array}{cc} \sum_{k=1}^{N_{b}}[ & -\mu_{k}\omega_{j}^{2}\int_{0}^{1}\phi_{k,i}\phi_{k,j}ds-\omega_{j}^{2}m_{k}^{*}\phi_{k,i}\left(1\right)\phi_{k,j}\left(1\right)-\alpha_{k}X_{i}\phi_{k,j}^{\prime \prime \prime }\left(0\right)+\alpha_{k}\int_{0}^{1}\phi_{k,i}^{\prime \prime }\phi_{k,j}^{\prime \prime }ds]\\ \; & -\mu \omega_{j}^{2}X_{i}X_{j}+K^{*}X_{i}X_{j}+X_{i}\sum_{k=1}^{N_{b}}\alpha_{k}\phi_{k,j}^{\prime \prime \prime }\left(0\right)=0\;, \end{array}$$
which gets simplified into
(21)$$\begin{array}{cc} -\omega_{j}^{2}[ & \sum_{k=1}^{N_{b}}\mu_{k}\int_{0}^{1}\phi_{k,i}\phi_{k,j}ds+\mu X_{i}X_{j}+\sum_{k=1}^{N_{b}}m_{k}^{*}\phi_{k,i}\left(1\right)\phi_{k,j}\left(1\right)] \end{array}$$
(22)$$\begin{array}{cc} \; & +\left[\sum_{k=1}^{N_{b}}\alpha_{k}\int_{0}^{1}\phi_{k,i}^{\prime \prime }\phi_{k,j}^{\prime \prime }ds+K^{*}X_{i}X_{j}\right]=0\;. \end{array}$$
so that the generalized mass and stiffness matrices can be defined as follows:(23)$$\begin{array}{cc} M_{ij}=[ & \sum_{k=1}^{N_{b}}\mu_{k}\int_{0}^{1}\phi_{k,i}\phi_{k,j}ds+\mu X_{i}X_{j}+\sum_{k=1}^{N_{b}}m_{k}^{*}\phi_{k,i}\left(1\right)\phi_{k,j}\left(1\right)] \end{array}$$(24)$$\begin{array}{cc} \; & K_{ij}=\left[\sum_{k=1}^{N_{b}}\alpha_{k}\int_{0}^{1}\phi_{k,i}^{\prime \prime }\phi_{k,j}^{\prime \prime }ds+K^{*}X_{i}X_{j}\right]\;. \end{array}$$
Note that the normalization condition is set to be $M_{jj}=1$, which yields
(25)$$\sum_{k=1}^{N_{b}}\mu_{k}\int_{0}^{1}\phi_{k,j}\phi_{k,j}ds+\mu X_{j}X_{j}+\sum_{k=1}^{N_{b}}m_{k}^{*}\phi_{k,j}\left(1\right)\phi_{k,j}\left(1\right)=1\;.$$

This allows to compute the coefficients $\left\{a_{1},\;\ldots ,\;a_{N_{b}}\right\}$ of the eigenvectors. In particular, the $\left(N_{b}-1\right)$ coefficients $\left\{a_{2},\;\ldots ,\;a_{N_{b}}\right\}$ are first obtained as function of $a_{1}$ through the system $A_{ij}a_{j}=0$ (for $i=2,\;\ldots ,\;N_{b}$), and finally $a_{1}$ is computed by imposing the previous normalization condition. According to this mass normalization, the frequencies are obtained as
(26)$$\begin{array}{c} \omega_{j}^{2}=K_{jj}=\sum_{k=1}^{N_{b}}\alpha_{k}\int_{0}^{1}\phi_{k,j}^{\prime \prime }\phi_{k,j}^{\prime \prime }ds+K^{*}X_{j}X_{j}\;. \end{array}$$
**Numerical results**. The mode shapes of the system, together with the corresponding natural frequencies, are computed using the linear equations of motion and solving the associated eigenvalue problem discussed in the previous section. Here and henceforth, we will consider $N_{b}=4$ microcantilevers attached to the shuttle mass.

The shuttle stiffness is set according to the formula $k=\omega^{2}\;M$, ω being a percentage of the lowest natural frequency of the sensors. The mechanical parameters are set to the following parameters: $k=10^{5}\;N/m$, M=40ρAl, $\rho =1236\;\mathrm{kg}/m^{3}$, h=b=50μm, $E_{M}=2.8\;\mathrm{GPa}$, $E_{C}=970\;\mathrm{GPa}$, $\nu_{M}=0.3$, $\nu_{C}=0.1$, l=270μm. Here, ρ is the mass density of the beam hosting matrix (i.e., epoxy), and *h* and *b* are the thickness and width of the microcantilevers cross section, respectively, so that A=hb is the area and $J_{22}=1/12\;bh^{3}$ is the second moment of area of the microbeam cross section about the $e_{2}$-axis (see  Figure 1).

The four microbeams are characterized by increasing volume fractions (from left- to right-beam), respectively $\phi_{C}=\left\{2,\;3,\;5,\;7\right\}\%$. The lowest 13 eigenvalues and the corresponding eigenmodes are depicted in  Figure 2; the 1st mode with $\omega_{1}=4.7423$ since it just involves the translational motion of the shuttle mass with no bending of the beams. The higher modes can be classified as local modes encompassing only the flexural motion of each cantilever separately. This indeed makes it possible to use the device as a multimass detector. In particular the 2nd mode, the first flexural of the first microcantilever, is associated with a frequency which is almost twice that of the 1st mode.

A parametric analysis is initially performed in order to investigate the sensitivity of the eigenfrequencies with respect to the CNT content variation and beam length. In  Figure 3 the first five frequencies are computed by varying the CNT volume fraction separately for each beam. Each curve shows that starting from certain values of the volume fraction, the five eigenfrequencies gain a plateaux, thus indicating a level of CNT content that does not affect the frequency properties of the system. This occurs for the shuttle mass eigenfrequency (i.e., the first one of the SISO system), for small values of CNT volume fraction (about 0.5%), for any beam considered. The same behaviour can be observed for each beam eigenfrequency, although this occurs for values of volume fraction $\phi_{C}$ larger and larger. In the same  Figure 3 we also indicate through dashed grid lines the values of $\phi_{C}$ selected for the next computations, thus proving to have an adequate frequency gap of the system.

The influence of the beam length is also considered in  Figure 4: the natural frequency of a beam made by a nanocomposite is compared with the natural frequency of a beam made of silicon in terms of the ratio $\omega_{\mathrm{cnt}}/\omega_{\mathrm{sil}}$ between the 1st natural frequency of the nanocomposite beam and that of the silicon beam, respectively. We explore in  Figure 4 how this ratio varies either by increasing the beam nominal length of 270μm by an amplification factor r∈[0,4], and by increasing the CNT volume fraction, starting from a nominal value of 2% then amplified again by *r*. While the natural frequency ratio monotically increases as the CNT volume fraction increases, by considering longer and longer lengths of CNT beam made of the same mixture, the natural frequency ratio tends to a plateaux. In particular, a nanocomposite beam with a 2.8% CNT volume fraction has the same value of the 1st natural frequency of a silicon beam of equal length.

If one chooses to set the first frequency of a microbeam to a certain target frequency, the microbeam could be designed in terms of a suitable length and CNT volume fraction.  Figure 5 shows indeed how to reduce the beam length as the CNT volume fractions increases. In particular, to have the same first frequency a silicon-made microbeam would be twice as long as a microbeam of nanocomposite with very low CNT volume fraction; a 8%-CNT nanocomposite beam has the same first frequency of a silicon microbeam of equal length.

## 4. Frequency Response Analysis

The Galerkin method is adopted to reduce the nonlinear integral-partial-differential equations of motion into a reduced-order model (ROM). By setting the unknown vector
(27)$$v{\left(s,t\right)}^{⊤}=\left[x\left(t\right),v_{1}\left(s,t\right),v_{2}\left(s,t\right),\ldots ,v_{N_{b}}\left(s,t\right)\right]\;,$$
the full system of equation can be cast as
(28)$$I\ddot{v}+Lv=-n\left(v,\dot{v},\ddot{v}\right)+f\;,$$
where I and L are the generalized inertia and stiffness operators, n is the vector collecting the nonlinear terms, and f is the vector of external forces. By expressing the solution as
(29)$$v{\left(s,t\right)}^{⊤}={\left[q_{1}\left(t\right)\;X_{1},\;\sum_{j=2}^{N_{m}}q_{j}\left(t\right)u_{j}\left(s\right)\right]}^{⊤}\;,$$
where we considered the *j*th mode $u_{j}^{⊤}=\left[X_{1},\phi_{k,j}\left(s\right)\right]$, for $k=1,\ldots N_{b}$, and substituting it into (Equation 28) yields
(30)$${\ddot{q}}_{j}Iu_{j}+q_{j}Lu_{j}=-n\left(q_{j}u_{j},{\dot{q}}_{j}u_{j},{\ddot{q}}_{j}u_{j}\right)+f\;,$$
where the summation convention has been adopted. Multiplying by $u_{i}^{⊤}$ and integrating over [0,1] yields the Galerkin discretized equations
(31)$${\ddot{q}}_{j}\int_{0}^{1}u_{i}^{⊤}Iu_{j}ds+q_{j}\int_{0}^{1}u_{i}^{⊤}Lu_{j}ds=-\int_{0}^{1}u_{i}^{⊤}n\left(q_{j}u_{j},{\dot{q}}_{j}u_{j},{\ddot{q}}_{j}u_{j}\right)ds+\int_{0}^{1}u_{i}^{⊤}fds\;,$$
which can be rewritten as
(32)$$M_{ij}{\ddot{q}}_{j}+K_{ij}q_{j}+n_{i}=f_{i}\cos\Omega t$$
with
(33)$$n_{i}=\int_{0}^{1}u_{i}^{⊤}n\left(q_{j}u_{j}\right)ds,\;f_{i}=\int_{0}^{1}u_{i}^{⊤}fds\;.$$

By taking one trial function for each cantilever, a 5-dof overall ROM is obtained (i.e., 1 dof for the shuttle mass and 1 dof for each of the four microcantilevers). This approach was shown to be acceptable in [53,54] where a convergence analysis was carried out. We will also consider the parameters $\zeta_{M}=0.01$ and ζ=0.011 for the shuttle mass damping ratio and the microcantilevers damping ratio, respectively; all the other mechanical parameters are those reported in the previous section.

In order to obtain the periodic responses and the frequency response curves, we employed the adaptive pseudo-arclength pathfollowing procedure proposed in [54], developed by some of the authors and freely available on the Zenodo repository website: https://doi.org/10.5281/zenodo.6616482 (accessed on 8 May 2023). In particular, the search of periodic solutions is based on computing the fixed points of the Poincaré map whose Jacobian turns out to be the monodromy matrix, whose eigenvalues are the Floquet multipliers which dictate the stability of the periodic solutions [55,56]. The Jacobian matrix of the Poincaré map is calculated according to a finite difference approach in state space, and it works as iteration matrix. The interested reader can find further details in [54].

The dynamic behavior of the device is investigated by computing the frequency response curves for different device configurations, in particular, accounting for the case of additional masses $m_{j}$ at the tip of the *j*th microcantilever; unless otherwise stated, we set for $m_{j}^{*}=0.01$, i.e., 1% of the microcantilever mass (ρAl is about 835 pg). Finally, we took as characteristic frequency, the first bending frequency of a microbeam made of neat polymer, so that $\omega_{c}=345,755$ rad/s.

In the following, we present and discuss some numerical results aimed at demonstrating the sensitivity of the SISO system to the presence of additional tip masses. Note that in all figures reporting the frequency response curves of the SISO system, thin curves are referred to systems without tip masses, thick curves to systems with tip masses.

The frequency response curves reported in Figure 6, Figure 7 and Figure 8 are recovered in a frequency bandwidth close to the frequencies of the lowest bending modes of the microbeams. We tested four SISO configurations: a system without additional masses, a system where a tip mass is added to the first (from left) microbeam, another system with tip masses at both the first and the third microbeam, and finally a system with tip masses deposited on all microbeams. Such figures show that the frequency shift is bounded within a narrow frequency bandwidth close to the frequency peak of the microbeams with mass absorbed at the tip. The comparison also proves that the sensitivity increases with the frequency of the considered local vibration mode, and it is in line with similar results reported in the literature obtained for simple microstructures (e.g., microcantilevers or clamped-clamped microbeams), see [57,58].

Similar qualitative results can be found in Figure 9, Figure 10 and Figure 11 which report frequency response curves for the same four SISO configurations when the excitation frequency is increased until nearing to the frequency of the second bending mode of the microbeams. The nonlinearities of the system entail a higher sensitivity than that predicted by the linear model, thus proving that nonlinear effects are advantageous to ensure higher performance of the MEMS mass sensor system. In fact, 1% deposited mass produces a shift of the peak about 0.125 in nondimensional frequency around the first mode, while such a value is amplified about by ten times around the second mode.

Such a mass sensitivity is further investigated for increasing values of additional masses. We assume to add the tip mass just to the first (from left) microbeam. Figure 12 and Figure 13 show the frequency response curves limited to tip deflection of such a beam, with excitation frequency ranging in values around the first and the second mode, respectively.

Such figures not only reveal the expected trend of the resonance peak shift upon increasing the tip mass, but they also highlight a change in the length of the unstable branch of the response curve. In fact, around the first bending mode frequency, the shift in frequency occurs with a hardening nonlinear response which becomes more and more evident (see Figure 12); on the other hand, around the second bending mode frequency, the softening nonlinear response does not change much qualitatively (see Figure 13).

## 5. Conclusions

In this work the operating principles of a nonlinear MEMS multi-mass sensor are investigated. The device consists of an array of microcantilevers, made of carbon nanotube nanocomposite material, possessing a functionalized surface at their tips that can absorb molecules of target materials. The microcantilever sensors are clamped onto a shuttle mass which, in turn, is subject to electrostatic excitation. The system is treated as a single input-single output (SISO) system. Each microcantilever is modeled as a nonlinear Euler-Bernoulli beam undergoing a moderately large resonance motions.

The modal characteristics of the mechanical system are investigated carrying out a parametric study varying the CNT volume fraction in each microbeam. The frequency response and stability are investigated by path following the periodic solutions of the device near the lowest three bending modes of the microcantilevers.

The main finding is that the dynamic mass sensitivity measured as frequency shifts induced by the target mass absorbed at the cantilevers tip is enhanced by the nonlinear resonances of the microcantilevers modes, especially for higher modes. In particular, around the first bending mode frequency, we gain a frequency shift of the 12% in frequency by adding a 10% tip-mass, while for the second mode, the frequency shift is of 9% with 5% additional mass.

Although the experimental validation of the proposed model has not been considered, and future works will focus on this point, the accuracy of the nonlinear beam model for CNT-nanocomposites, adapted to the specific SISO system here theoretically designed, was widely validated in previous research contributions. In the present work, we showed how the fine tuning of the frequencies of the microcantilevers is positively effected by the carbon nanotube volume fraction in the nanocomposite material. Such innovative potentiality paves the way towards an effective design of multimass sensors since the microcantilevers may self-detect their motion exploiting the electrical conductivity of the CNT nanocomposites.

## Figures and Tables

**Figure 1 nanomaterials-13-01808-f001:**
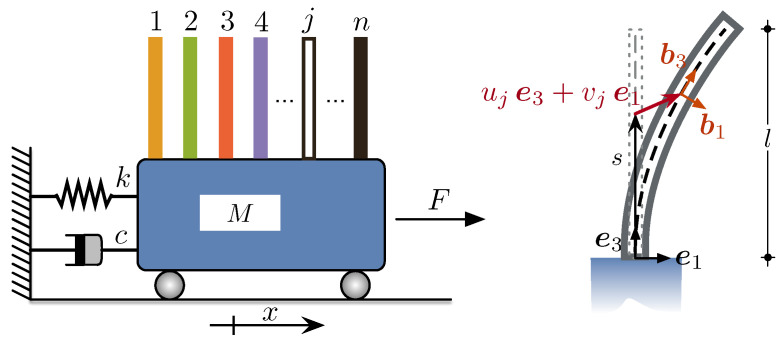
Array of nanocomposite microcantilevers clamped onto the shuttle mass. The microcantilevers are numbered from left to right. The coordinate *x* denotes the shuttle mass displacement, *F* the amplitude of harmonic direct excitation; on the right, a schematic representation of the deflection of the *j*th microcantilever together with the fixed and local frames and the coordinate *s* along the undeformed centerline.

**Figure 2 nanomaterials-13-01808-f002:**
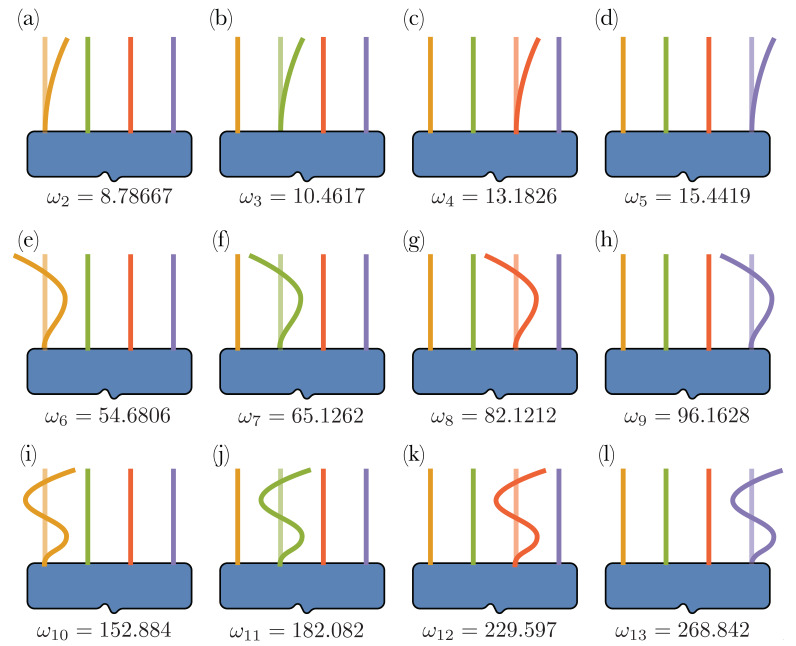
Mode shapes of the device, from the 2nd to the 13rd mode. The 1st mode with $\omega_{1}=4.7423$, involving only the translational motion of the shuttle mass with no bending of the beams, is here omitted. Parts (**a**–**d**) depict the first flexural beam modes associated with 1st–4th microcantilever, respectively, (**e**–**h**) the second flexural mode, and (**i**–**l**) the third flexural mode.

**Figure 3 nanomaterials-13-01808-f003:**
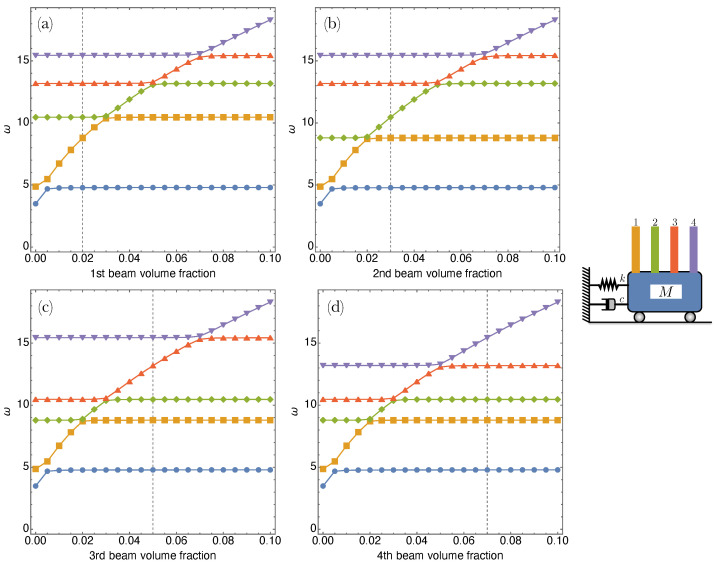
Device frequencies (from the first up to the fifth mode) varying the CNT volume fraction of each microcantilever. Dashed grid lines indicate the opted values in the analyses for the chosen volume fractions for the subsequent analyses. Microcantilevers ordered as in the picture on the right, with (**a**–**d**) plots corresponding to CNT volume fraction variations of the 1st–4th microcantilever, respectively.

**Figure 4 nanomaterials-13-01808-f004:**
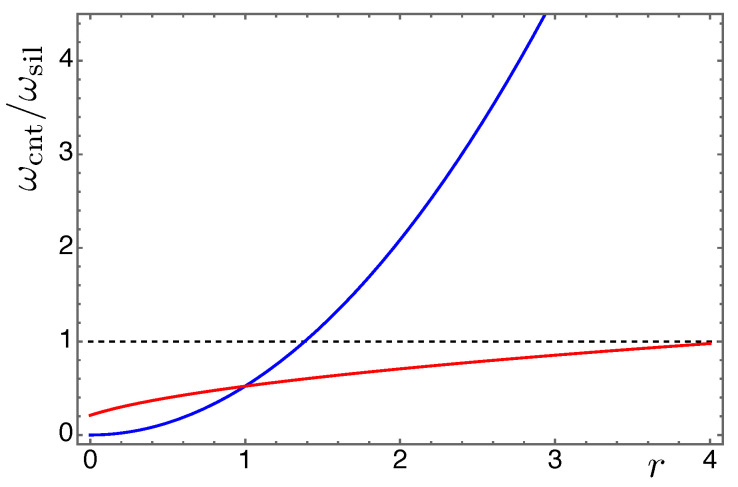
Natural frequencies ratio of a beam made by a nanocomposite ($\omega_{\mathrm{cnt}}$) with respect to a beam made of silicon ($\omega_{\mathrm{sil}}$). Ratio varying for increasing beam length as r270μm of the silicon beam (blue curve), and ratio varying for increasing volume fraction as r0.02 of the CNT beam (red curve).

**Figure 5 nanomaterials-13-01808-f005:**
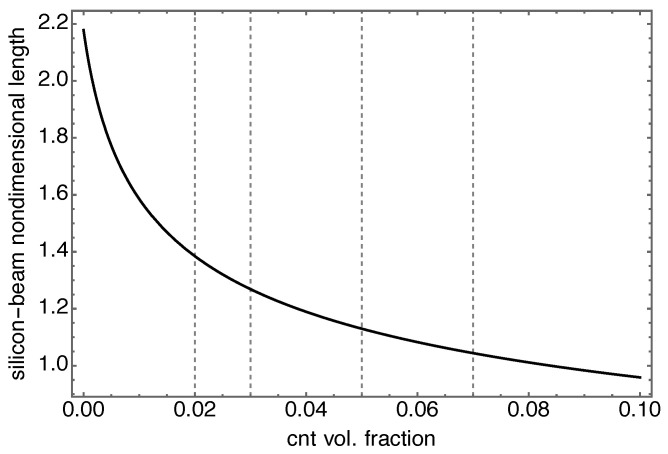
Dependency of silicon-beam length on nanocomposite-beam volume fraction to have the same first beam natural frequency. The silicon-beam length is adimenzionalized with respect to the nanocomposite-beam length equal to 270μm. Gridlines denote the volume fraction values used in the device.

**Figure 6 nanomaterials-13-01808-f006:**
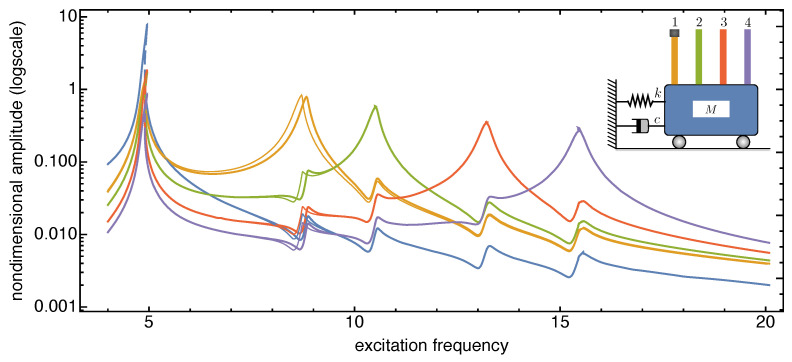
Frequency response curves of the device with and without tip mass when 1% added mass is absorbed onto the first microcantilever tip (from left). The solid (dashed) curves indicate stable (unstable) periodic responses.

**Figure 7 nanomaterials-13-01808-f007:**
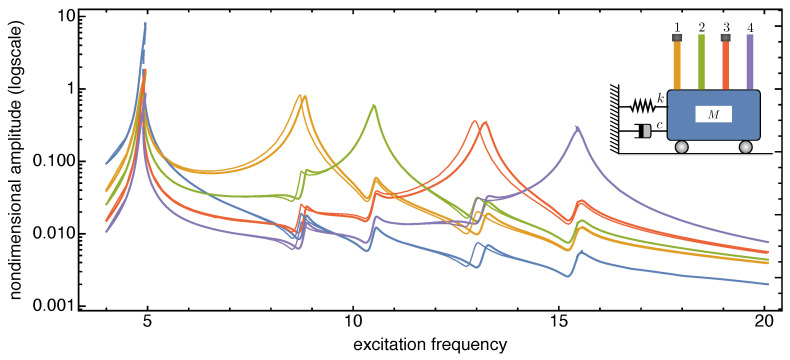
Frequency response curves of the device with and without tip mass when 1% added mass is absorbed onto both the tips of the first and third microcantilevers (from left). The solid (dashed) curves indicate stable (unstable) periodic responses.

**Figure 8 nanomaterials-13-01808-f008:**
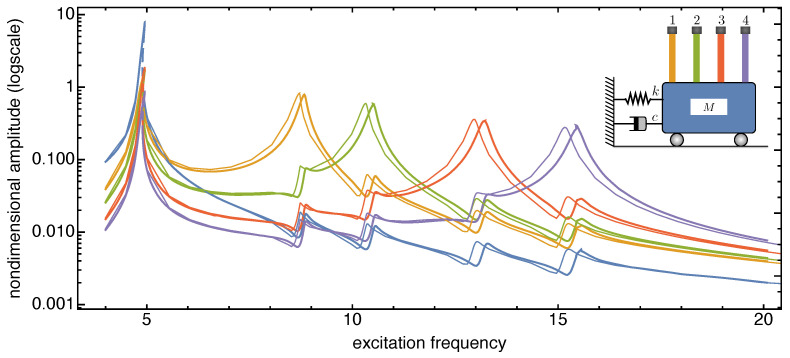
Frequency response curves of the device with and without tip mass when 1% added mass is absorbed onto all the microcantilevers tips. The solid (dashed) curves indicate stable (unstable) periodic responses.

**Figure 9 nanomaterials-13-01808-f009:**
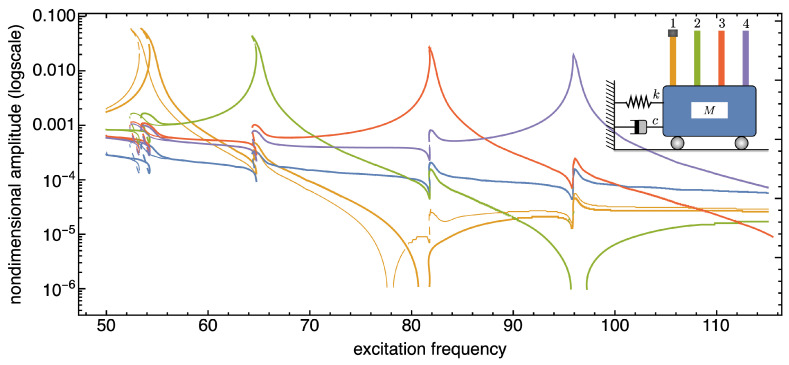
Frequency response curves of the device with and without tip mass when 1% added mass is absorbed onto the first microcantilever tip (from left). Frequency bandwidth restricted to the second bending modes of the microcantilevers. The solid (dashed) curves indicate stable (unstable) periodic responses.

**Figure 10 nanomaterials-13-01808-f010:**
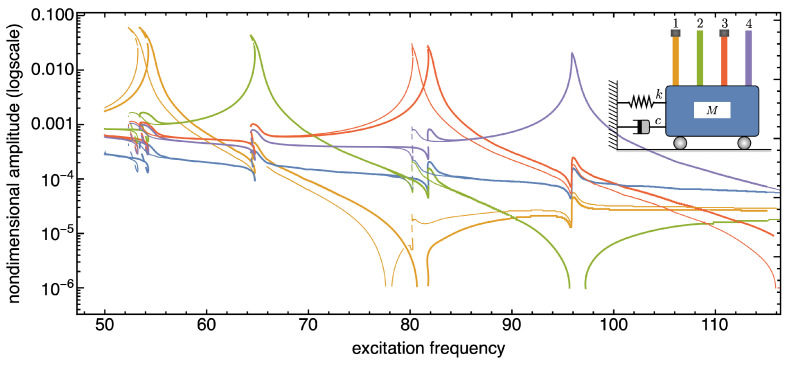
Frequency response curves of the device with and without tip mass when 1% added mass is absorbed onto both the tips of the first and third microcantilevers (from left). Frequency bandwidth restricted to the second bending modes of the microcantilevers. The solid (dashed) curves indicate stable (unstable) periodic responses.

**Figure 11 nanomaterials-13-01808-f011:**
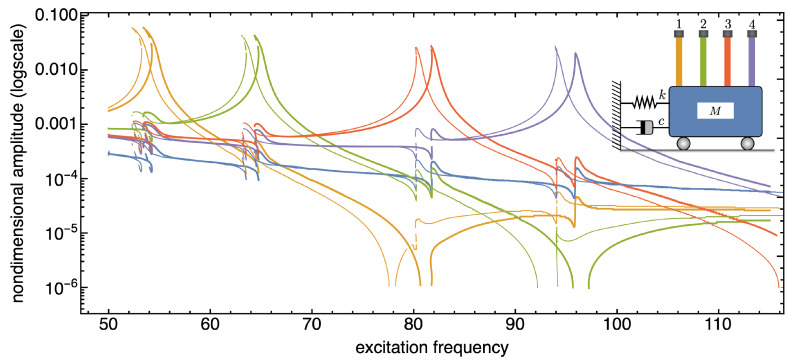
Frequency response curves of the device with and without tip mass when 1% added mass is absorbed onto all the microcantilevers tips. Frequency bandwidth restricted to the second bending modes of the microcantilevers. The solid (dashed) curves indicate stable (unstable) periodic responses.

**Figure 12 nanomaterials-13-01808-f012:**
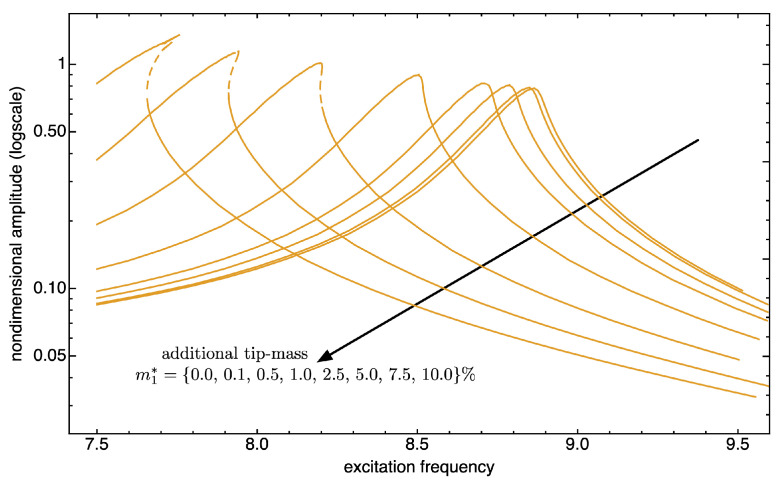
Absorbed mass sensitivity. Frequency response curves of the first bending mode of the first cantilever with an additional mass at the tip. See also Figure 6. The solid (dashed) curves indicate stable (unstable) periodic responses.

**Figure 13 nanomaterials-13-01808-f013:**
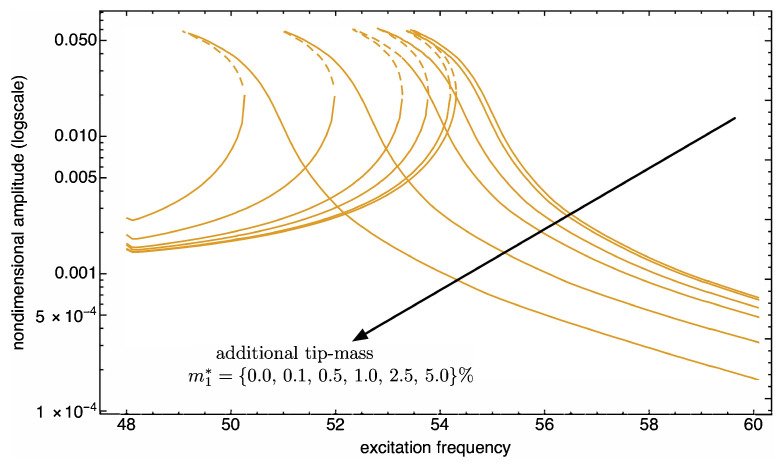
Absorbed mass sensitivity. Frequency response curves of the second bending mode of the first cantilever with an additional mass at the tip. See also Figure 9. The solid (dashed) curves indicate stable (unstable) periodic responses.

## Data Availability

Not applicable.

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
