# Peer review of "Nonlinear Dynamic Response of Nanocomposite Microbeams Array for Multiple Mass Sensing"

_nanomaterials, 2023, doi:10.3390/nano13111808_

Round 1

Reviewer 1 Report

In this article, the authors considered a nonlinear MEMS multimass sensor. The MEMS design consists of an array of nonlinear microcantilevers clamped to a shuttle mass which, in turn, is attached to a linear spring and a dashpot. The microcantilevers are made of a polymeric hosting matrix reinforced by aligned carbon nanotubes (CNT) with different CNT volume fractions in each cantilever so as to suitably tune the frequency bandwidth of the device. The advantages of a single input-single output (SISO) system are here exploited for the design, actuation and detection principles as in previous literature.

The paper is well written, I recommended it for publication after considering the following points:

1.      The abstract will define in one paragraph and should be short.

2.      Add the list of nomenclature in order to easy for the readers.

3.      The introduction section needs modification with new results of nanomaterials. You can consider the recent results with nanoparticles “Linear and quadratic convection on 3D flow with transpiration and hybrid nanoparticles; Nonlinear movements of axisymmetric ternary hybrid nanofluids in a thermally radiated expanding or contracting permeable Darcy Walls with different shapes and densities: Simple linear regression”.

4.      Give references to equations.

5.      Add a comma and dot in the appropriate place after each equation.

Minor editing of English language required

Reviewer 2 Report

I have read the manuscript titled: Nonlinear dynamic response of nanocomposite microbeams array for multiple mass sensing. It has been well-written and readable. The title is interesting and novel. The performed science is worth studying. The manuscript can be recommended for publication, however not in the current format. The following comments are required to be addressed,

*What is the exact application of such a modeled system?

*What about the validation of the results with experiments?

*Why the microscale effects have not been considered? For example, strain gradient or couple stress effects? 

Without considering the size effect, you did not analyze a microbeam.

*As I see in Fig.2, the mode shapes are normalized for each microbeam and the micro-cantilevers vibrate independently. What about a mixed mode?

Reviewer 3 Report

1. Need to revise Abstract. Add numerical values of the results obtained. Give information on mathematical modeling. Clarify what type of research is used : theoretical or experimental?

2. Supplement the literature review. Especially expand the information on carbon nanotubes. In particular we can add some information on synthesis of CNT: Shchegolkov A.V., Shchegolkov A.V. Synthesis of carbon nanotubes using microwave radiation: technology, properties and structure. Russ J Gen Chem 92, 1168-1172 (2022). https://link.springer.com/article/10.1134/S1070363222060329

Nanotechnologies and the distribution peculiarities of CNT in polymeric matrices:

Shchegolkov, A.V.; Jang, S.-H.; Shchegolkov, A.V.; Rodionov, Y.V.; Glivenkova, O.A. Multistage Mechanical Activation of Multilayer Carbon Nanotubes in Creation of Electric Heaters with Self-Regulating Temperature. Materials 2021, 14, 4654. https://doi.org/10.3390/ma14164654.

3. Formulate a research goal and add research objectives: 1,2,3....

4. Introduce divisive notations in the figures : a, b, c ......

5. Supplement the conclusions of the research with numerical values of key parameters.

Moderate editing of English language
